# Study on Demulsification Technology of Heavy Oil Blended in Xinjiang Oilfield

Jungang Zou [1], Yaermaimaiti Patiguli [1], Jun Chen [1], Awan Alimila [1], Bin Zhao [1] and Junwei Hou [2,*]

[1] Xinjiang Oilfield Company, Karamay 834000, China
[2] State Key Laboratory of Heavy Oil Processing, China University of Petroleum—Beijing at Karamay, Karamay 834000, China
[*] Correspondence: junweihou@cupk.edu.cn

**Abstract:** HYW (Hong Yi Wu line) heavy oil emulsion in Xinjiang Oilfield (Karamay, China) is a kind of heavy oil with high viscosity and high emulsification. Its viscosity reaches 120,000 mPa·s at 40 °C. The emulsion has no demulsification. Even if the demulsification temperature reaches 90 degrees, the concentration of demulsifier reaches 260 mg/L. In this paper, a new process of thermochemical demulsification of heavy oil after blending is studied. First, SE low-viscosity oil with viscosity of 640 mPa·s and water cut of 90% was selected as blended oil. Study the viscosity of SE line and HYW line at different temperatures after fully blended. The results show that the heavy oil blended model conforms to Bingham model. When the temperature is 40 °C and the content of SE line is 30%, the viscosity is less than 10,000 mPa·s. With the increase of temperature, the viscosity continues to decline. When the temperature exceeds 80 °C, the viscosity is less than 1000 mPa·s. The final design SE line content is 30%, the demulsification temperature is 70 °C, and the demulsifier concentration is 160 mg/L as the best demulsification parameter. The field results show that the demulsification rate of heavy oil in this process reaches more than 90%. This experiment lays a foundation for demulsification of high emulsified crude oil developed by heavy oil in Xinjiang oilfield.

**Keywords:** heave oil blended; process optimization; demulsification; temperature

## 1. Introduction

Heavy oil refers to high-viscosity heavy crude oil with degassing viscosity of 1000–100,000 mPa·s (millipascals per second) at reservoir temperature. In addition to high viscosity, heavy oil also has high density. Heavy oil is an important refining and chemical raw material with less light fraction and high content of resin and asphalt [1–3].

Heavy oil has high viscosity and poor fluidity, and water drive technology cannot be used to develop heavy oil. At present, heavy oil development mainly uses steam flooding, fire flooding, $CO_2$ flooding, chemical flooding and other technologies [4–7]. Colloid and asphaltene are polar substances in crude oil with high interfacial activity. The content of colloid and asphaltene in heavy oil is much higher than that in light oil, and it is easier to form emulsions. In addition, the viscosity of heavy oil is large, and it is easier to carry small inorganic particles in the formation, which will be adsorbed at the oil–water interface. Therefore, when heavy oil migrates in the formation and tubing, it will have strong interaction with the surrounding media and will produce "Picking" emulsion with very strong stability and high viscosity [7,8].

HYW heavy oil emulsion in Xinjiang Oilfield is a typical Pickering emulsion. Its viscosity reaches 120,000 mPa·s at 40 °C, and the water content is 55%. When the temperature rises to 90 °C and the concentration of demulsifier reaches 260 mg/L, the demulsification effect is still poor. The emulsion with high viscosity and high-water content is accumulated in the gathering station, which has a great impact on the subsequent production. Therefore, it is urgent to study a demulsification method for heavy oil emulsion [9–13].

At present, the common demulsification technologies are as follows:

(1) Thermochemical sedimentation dehydration, which is a dehydration process that heats the water containing crude oil or crude oil produced fluid to a preset temperature and adds a proper amount of chemical demulsifier [14]. Zhang et al. [15] studied the demulsification effect of clear water type, polyether type and multi-ethylene-polyamine type demulsifiers on a weak alkali ASP flooding emulsion in Daqing Oilfield under thermochemical heating conditions. The results showed that the three kinds of demulsifiers have different degrees of demulsification effect on weak base ASP flooding under heating conditions, and the demulsification and dehydration rate of 100# (polyether type) demulsifier among the three kinds of demulsifiers is faster. The higher the temperature is, the higher the demulsification rate is, and the better the separation effect of the emulsion is. When the temperature rises to 55 °C, the extreme temperature point appears, and the demulsibility of the emulsion is the best. When the temperature continues to rise, the emulsion emulsifies. The 100# polyether type demulsifier has good demulsification effect in the oil phase and water phase of 10 mg·L-1, which is suitable for demulsification of produced liquid of weak base ASP flooding in a block of Daqing Oilfield. Yang et al. [16] studied the effects of heating temperature and demulsification dosage on the demulsification effect and demulsification rate of Dagang oilfield emulsion. The test results show that for a certain amount of crude oil emulsion, the dosage of demulsifier is at an optimum value; temperature will affect the optimum value of demulsifier. With the increase of temperature, the optimum value of demulsifier dosage used in demulsification of crude oil emulsion will decrease; at the same time, the amount of demulsification added has a certain effect on the dehydration rate of the emulsion.

(2) Electrochemical demulsification technology mainly uses a high-voltage pulsed electric field for demulsification. First, the double electric layer of the emulsion is destroyed. After that, charged droplets are arranged into chains, and finally polymerized and demulsified. Therefore, electrochemical demulsification is mainly applied to the demulsification of oil-in-water emulsion with high conductivity [17]. He et al. [18] studied the effect of high-frequency electric pulse dehydration technology on the demulsification of Bohai polymer-containing crude oil emulsion. The results showed that (i) raising the dehydration temperature for polymer-containing emulsified oil can greatly improve the electric dehydration efficiency. (ii) When the temperature is higher than 90 °C, the water content is not less than 10%, and the high-frequency electric pulse is used for electric dehydration, the electric dehydration can achieve a relatively ideal effect. (iii) When high frequency electric pulse is used for dehydration process, proper demulsifier can be added to improve dehydration rate. Pan et al. [19] studied the effect of phase inversion on electrostatic coalescence and dehydration by using the static crude oil electrostatic coalesce experimental device. The effects of electric field intensity and frequency on dehydration were also studied in the case of aging oil emulsion ware passing through the high frequency and voltage electric field by using a specific electrostatic coalesce experimental device. The results showed that the water content at the phase inversion point was about 40%, the viscosity of aging oil emulsion increased obviously during phase inversion process, which hindered the dehydration. The dehydration efficiency was up to 97.8% under the optimal conditions of electric field intensity 1.25 kV/cm and frequency 2.5 kHz when the water content was 30%, the dehydration efficiency was only 4.2% under the same electric field parameters except 50 Hz frequency. A high-frequency electric field was more effective than a regular electric field.

(3) Centrifuge dehydration, the use of centrifugal separation technology to promote the demulsification and dehydration of crude oil emulsion, which has been widely used in Shengli Oilfield [20]; Liu et al. [21] of Southwest Petroleum University studied the effect of types of demulsifier, demulsifier dosage, types of demulsification additives, demulsification additive dosage, demulsification temperature, emulsification time, centrifugal velocity on dewatering efficiency, and the de-oiling rate of oil tank bottom

sludge. The best demulsifier was WDP-9, based on a number of experiments, and was a compound of polyaluminium chloride and polyacrylamide. The results showed that the dewatering efficiency and de-oiling rate of oil tank bottom sludge were 85.70% and 67.10%, respectively. The optimum parameters were as follows: WDP-9 concentration of 500 mg·L$^{-1}$, polyaluminium chloride concentration of 75 mg·L$^{-1}$, polyacrylamide concentration of 75 mg·L$^{-1}$, demulsification temperature of 60 °C, demulsification time of 2 h, centrifugal velocity of 10,000 r·min$^{-1}$, and centrifugation two times, for 10 min each time.

(4) Ultrasonic dehydration, crude oil emulsion is subject to mechanical, thermal and cavitation effects of ultrasound. The mechanical action can promote the mechanical vibration of small water droplets and accelerate the collision and coalescence of water droplets; an ultrasonic wave carries energy in the form of ripples and spreads in the emulsion. The emulsion continuously absorbs the vibration energy carried by it and converts it into heat energy, which can increase the temperature of the emulsion and is conducive to demulsification and dehydration [22,23]. Yu et al. [24] studied the effect of ultrasonic demulsification technology on demulsification. The results show that the ultrasonic wave plays a positive role in demulsifier dispersion and demulsification and can improve the oil–water separation efficiency under the temperature of well discharge liquid with a good water separation effect. At 40 °C, the water content of heavy oil produced fluid (density 0.965 kg/m$^3$, viscosity in working condition 2900 mPa·s, water content of emulsified oil 39%) can be reduced from 91% to below 25% by ultrasonic demulsification treatment, and the subsequent heating energy consumption can be reduced by more than 50%. The application of ultrasonic technology in oil–water separation can reduce the energy consumption and operation cost, which has good development potential. However, if the time of ultrasonic action is too long or the intensity is too large, it will present a negative effect on demulsification. The operating parameter should be adjusted according to specific oil properties.

(5) Two stage dehydration process. Fu et al. [25] studied the demulsification and dehydration process coupled with ultrasonic and centrifugal technology. The results show that when the centrifugal time is 20 min, the centrifugal speed is 4000 r·min$^{-1}$, the ultrasonic irradiation time is 45 min, the ultrasonic settling time is 120 min, the ultrasonic power is 300 W, and the reaction temperature is 50 °C, the optimum deoiling rate is 92.46%. Bo et al. [26] studied the demulsification and dehydration process coupled with ultrasonic and electrochemical technology. The results showed that the action time of ultrasound, the action sequence of ultrasound and electric field, the intensity and frequency of ultrasound have great influence on the dehydration effect. When the combined action time of electric field and ultrasonic field is 5 min and 10 min, the final dehydration effect is better. When the ultrasonic intensity is lower than the critical value, the result is demulsification. When the ultrasonic intensity exceeds the critical value, the result of ultrasonic irradiation is emulsification. The viscosity and dehydration rate of crude oil both increase first and then decrease with temperature, and the combined acoustic and electric fields are more suitable for treating crude oil emulsions with an initial water content of 15–25%. Ni et al. [27] studied the demulsification process of solar light, heat, electric single field and photo-thermal-electric composite field. Firstly, the effects of oil content, alkali concentration, surfactant concentration, polymer concentration, emulsification shear rate and emulsification time on the stability of ASP flooding emulsion were investigated by single factor experiment. The results showed that the stability of emulsion was mainly determined by viscosity, interfacial tension, interfacial facial mask strength and other factors. Secondly, solar light, thermal, electric single field and composite field demulsification experiments were carried out respectively. The changes of viscosity, particle size and distribution, Zeta potential and interfacial tension of the emulsion were measured in real time. The results show that the photo-thermal-electric composite field treatment process achieves

good demulsification effect on the ternary flooding emulsion by reducing viscosity, reducing surface charge density and increasing interfacial tension. Thirdly, the oil content of emulsion water layer after single field and compound field demulsification experiments was measured by UV–Vis spectrophotometry. The results show that the photo-thermal-electric three-field synergy has the best demulsification effect. Finally, it is concluded that the demulsification mechanism of solar photo-thermo-electric process mainly includes photo-thermo-chemical degradation, viscosity reduction, electrochemical oxidation, air flotation effect, dipole coalescence and electrophoretic coalescence. Demulsification treatment of produced water from ASP flooding is an important part of the application and development of ASP flooding technology.

Different demulsification technologies and their characteristics are shown in Table 1. However, these technologies are difficult to implement in the HYW heavy oil emulsion of Xinjiang Oilfield. First of all, the amount of HYW emulsion is large, more than 5000 tons per day, and the temperature is raised to more than 90 °C for demulsification, so the energy consumption is too high. Second, the emulsion is water-in-oil type, with low conductivity, and cannot use electrochemical demulsification technology. Thirdly, the emulsion has strong adhesion and will adhere to the centrifuge, so the centrifuge technology cannot be used. Finally, the viscosity of the emulsion is too high, and the effective range of ultrasonic cavitation is too small to be used on a large scale. Therefore, it is very important to study a new heavy oil demulsification technology with low price, low energy consumption and simple technology.

**Table 1.** Different demulsification technologies and their characteristics.

| Demulsification Technology | Region of Use | Characteristic |
|---|---|---|
| Blended oil | Xinjiang Oilfield, Shengli Field, Liaohe Oilfield | Narrow application range and high control difficulty |
| Thermochemistry | Most oil fields | Excessive energy consumption |
| Ultrasonic | Saudi Aramco Oilfield; Bohai Oilfield | Narrow application range and high energy consumption |
| Microwave | Laboratory stage | Narrow application range and high energy consumption |
| Biotechnology | Huabei Oilfield, Daqing Oilfield | Narrow application range and biological resistance |
| Electrochemistry | Daqing Oilfield, Xinjiang Oilfield, Nanhai Oilfield | Narrow application range and high energy consumption |
| Centrifugal technology | Ansai Oilfield, Caofeidian Oilfield, Dagang Oilfield | Narrow application range and high energy consumption |
| Evaporation technology | Liaohe Oilfield | Low energy consumption but high time cost |

The blended of heavy oil and light oil can greatly reduce the viscosity of heavy oil, which is conducive to the pipeline transportation of heavy oil [28,29]. However, the technology of demulsification by blended has not been reported yet. In this paper, the SE emulsion with viscosity of 640 mPa·s was selected firstly, and then the effects of blended ratio and temperature on viscosity, density, emulsion size, interfacial tension, Zeta potential and water separation rate were studied. The results show that when the temperature is 70 °C and the proportion of HYW is 70%, the 2-h water separation rate can reach 90%.

## 2. Materials and Methods

### 2.1. Material and Reagent

Produced liquid of heavy oil in HYW line (No. 1 Oil Production Plant of Xinjiang Oilfield); Produced fluid of heavy oil in SE line (No. 1 Oil Production Plant of Xinjiang Oilfield); A6 type cationic demulsifier for heavy oil treatment station (industrial product, Xinjiang Keli Company, Karamay, China).

### 2.2. Experimental Method

According to the Einstein–Stokes equation [30], the speed of oil–water separation is proportional to the density difference between emulsion and water, the size of emulsion and the viscosity; at the same time, demulsification of emulsion is related to viscosity, temperature, interfacial tension and absolute value of Zeta potential.

Firstly, the viscosity temperature curves at different blended ratios are measured by using Physica MCR301 rheometer. The temperature was 40 °C, 50 °C, 60 °C, 70 °C, 80 °C, 90 °C, and the rotational speed was 1 s$^{-1}$. Secondly, the size and type of emulsion are measured using Zeiss AXIOSKOP 40 microscope. The emulsion density is measured by XFCNMD-320API. The emulsion interfacial tension is measured with TX500C interfacial tensiometer, with the rotating speed of 6000 rpm and temperature of 70 °C. Zeta potential of emulsion is measured with Malvern zetasizer nano at 70 °C. The blended ratio is 10:0, 9:1, 8:2, 7:3, 6:4, 5:5, 4:6, 3:7 and 0:10 respectively. The test process is shown in Figure 1.

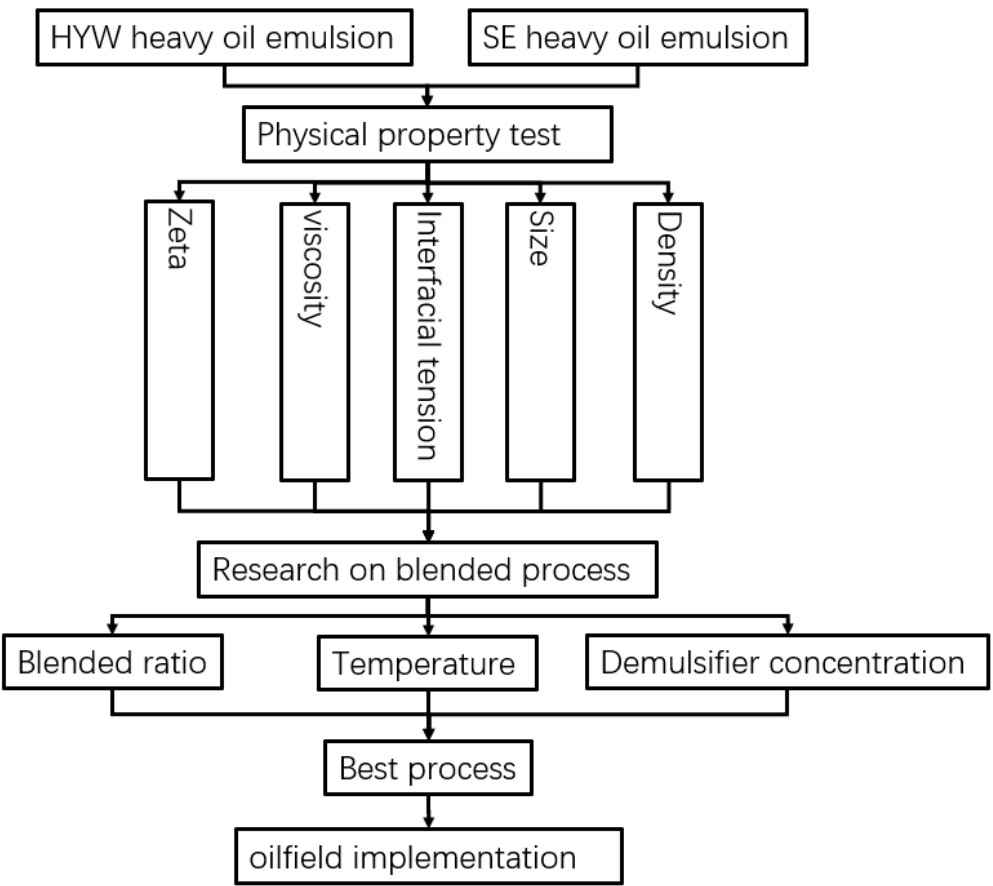

**Figure 1.** The test flow chart.

The water separation rate of emulsion was measured using a 100 mL conical test tube and a constant temperature water bath. Add demulsifiers of different concentrations, adjust different blended ratios, measure the volume $V_1$ of the separated water after 2 h,

and measure the water content $V_2$ of the mixed emulsion with a water content meter, then the water content meets the following Equation (1).

$$c = \frac{V_1}{V_2} \times 100\% \tag{1}$$

### 2.3. Analysis Method

The model and country of the experimental instrument are shown in Table 2.

**Table 2.** Instrument model and country.

| Name | Type | Manufacturer | Place of Origin |
|---|---|---|---|
| Rheometer | MCR301 | Antonpa | Austria |
| Constant temperature water bath | HH-601A | Kanglu | China |
| Densitometer | DA-300PF | Hongtuo Instrument | China |
| Microscope | AXIOSKOP 40 | Carl Zeiss Optics | Germany |
| Zeta potentiometer | Nano | Malvern | England |
| Interface tensiometer | TX-500C | Kono Industries Co., Ltd. | America |

## 3. Results

### 3.1. Basic Parameters

The basic physical properties of HYW and SE emulsions are shown in Table 3.

**Table 3.** HYW and SE emulsion characteristics.

| Name | Density g/cm$^3$ | Viscosity mPa·s | Emulsion Size Mm | Emulsion Type | Interfacial Tension mN/m | Zeta Potential mV |
|---|---|---|---|---|---|---|
| HYW | 0.99 | 120,000 | 5 | W/O | 6 | −105 |
| SE | 0.92 | 640 | 40 | O/W | 30 | −60 |

It can be seen that the initial viscosity of HYW line emulsion is 120,000 mPa·s, and the density reaches 0.99 g/mL, which is close to the density of water; The size of emulsion is less than 5 microns. The interfacial tension is only 6 mN/m, and the absolute value of Zeta potential is as high as 105 mV. According to the Einstein–Stokes equation, the water separation speed of the emulsion is proportional to the density and the square of the size of the emulsion, and inversely proportional to the viscosity, so the water separation is very slow. According to the DLVO theory, emulsion demulsification is related to the interfacial tension and Zeta. The lower the interfacial tension, the easier it is to emulsify. The smaller the interfacial tension, the stronger the emulsification, and the higher the absolute value of Zeta potential, the more stable the emulsion. Therefore, HYW emulsion is difficult to demulsify.

The ion distribution of water in the emulsion is shown in Table 4. It can be seen that the concentration of chloride ion and sodium ion in HYW line is high, and the total salinity is 5600 mg/L; the concentration of calcium ion in SE line is relatively high, and the total mineralization is 5500 mg/L. The difference between the two kinds of water is very small.

**Table 4.** Ion analysis of water contained in HYW and SE emulsion.

| Name | Cl$^-$ mg/L | Na$^+$/K$^+$ mg/L | SO$_4^{2-}$ mg/L | OH$^-$ mg/L | CO$_3^{2-}$ mg/L | HCO$_3^-$ mg/L | Ca$^{2+}$ mg/L |
|---|---|---|---|---|---|---|---|
| HYW | 3072 | 2359 | 97.4 | 0 | 0 | 987 | 44.1 |
| SE | 2707 | 1861 | 17.1 | 0 | 0 | 771.78 | 152.4 |

### 3.2. Blended Ratio, Temperature and Viscosity Distribution Diagram

Figure 2 and Table 5 show the viscosity distribution of HYW and SE emulsion at different temperatures after blended in different proportions. It can be seen that at 40 °C, when SE emulsion accounts for 30% of the total, the viscosity can be reduced to 10,000 mPa·s; When the temperature rises to 70 °C, the SE emulsion content reaches 50%, the viscosity can be reduced to 2000 mPa·s. The mixed viscosity model of heavy oil meets Bingham model [31–33].

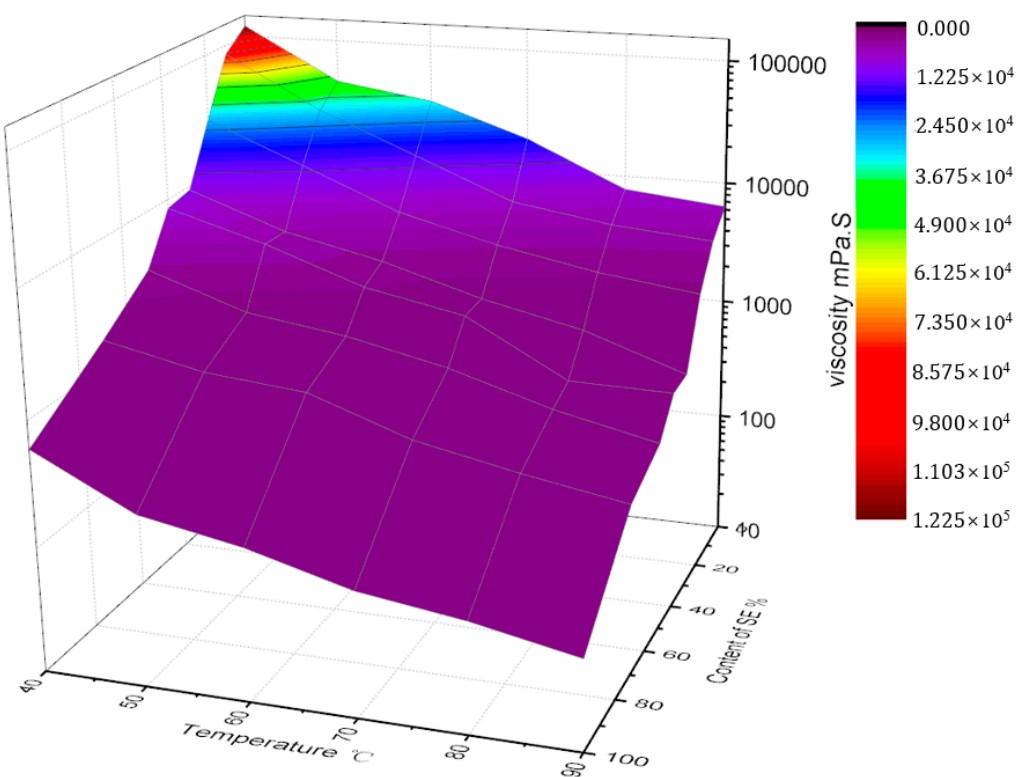

**Figure 2.** Viscosity distribution of HYW and SE mixed in different proportions at different temperatures.

**Table 5.** Viscosity distribution of HYW and SE mixed in different proportions at different temperatures.

| Content of SE Emulsion % | 40 °C | 50 °C | 60 °C | 70 °C | 80 °C | 90 °C |
|---|---|---|---|---|---|---|
| 0 | 122,333 | 45,122 | 33,989 | 18,260 | 7777 | 6368 |
| 10 | 84,467 | 31,418 | 14,643 | 6804 | 5027 | 4255 |
| 20 | 28,694 | 13,471 | 6227 | 3821 | 2648 | 2059 |
| 30 | 9353 | 4794 | 3153 | 1805 | 1179 | 611 |
| 50 | 8438 | 4933 | 2551 | 1752 | 619 | 597 |
| 70 | 3390 | 1786 | 1361 | 898 | 506 | 331 |
| 100 | 1627 | 1084 | 908 | 468 | 317 | 228 |

### 3.3. Density Curve

The density of HYW and SE emulsion was measured with a densimeter. As shown in Figure 3, the density of HYW emulsion was 0.99 g/cm$^3$, while that of SE emulsion was 0.92 g/cm$^3$. The higher the content of HYW emulsion, the greater the density; the higher the temperature, the lower the density. There are two main reasons: first, after the temperature rises, part of the emulsion begins to break. Under the action of gravity, the water with high density sinks, and the oil with low density floats, resulting in the decrease of the density of the water-in-oil emulsion. Second, the liquid has the characteristics of thermal expansion and contraction. When the temperature increases, the volume increases and the density decreases.

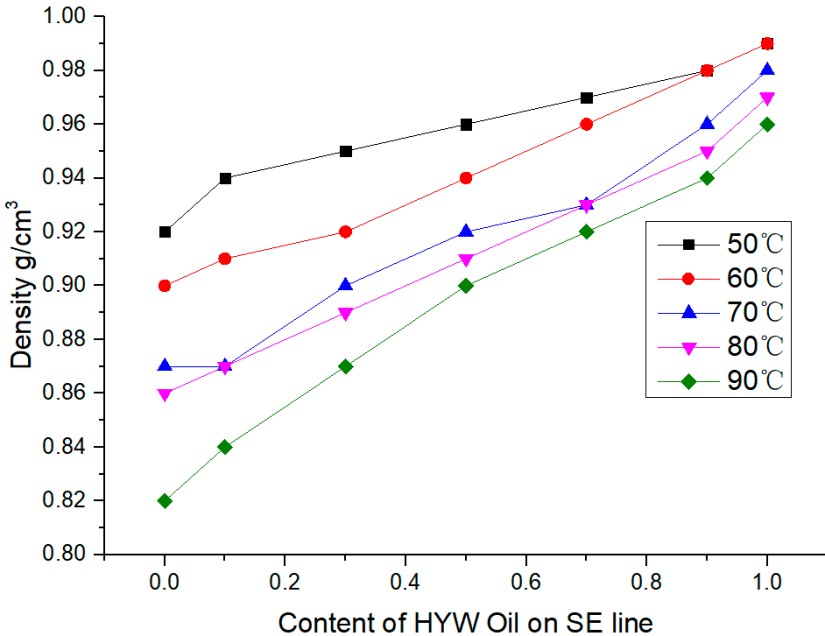

**Figure 3.** The impact of current density on density.

### 3.4. Size and Type of Emulsion

Figure 4 shows the optical microscope photos from HYW emulsion and SE emulsion. It can be seen that the HYW emulsion is mainly water-in-oil emulsion with a diameter of 10–20 μm, average particle size 15 μm. The SE emulsion is mainly oil-in-water emulsion with a diameter of 50–100 μm, average diameter 60 μm. After blended, the mixed solution generally presents oil-in-water emulsion, and there is a small amount of water-in-oil-in-water emulsion with the diameter of 100 μm. When the content of HYW emulsion is 90%, it is a water in oil emulsion. When the content of HYW is less than 70%, water-in-oil emulsion becomes an oil-in-water emulsion. The average diameter of the emulsion is more than 120 μm. With the continuous decrease of the content of HYW, the size of the emulsion slowly decreases. Therefore, the content of HYW emulsion is 70%, the blended emulsion has the largest size.

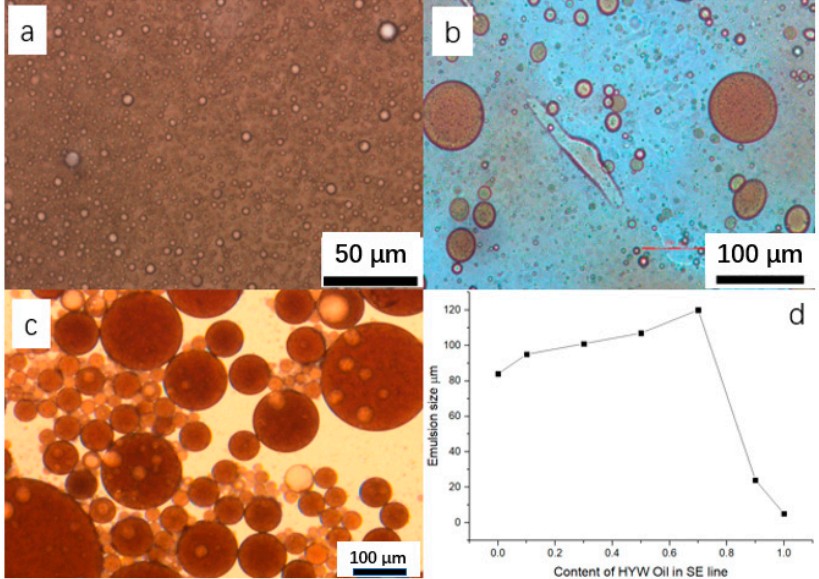

**Figure 4.** (**a**). The optical photos of HYW emulsion; (**b**). SE emulsion; (**c**). HYW mixed with 70% emulsion; (**d**). Size distribution of different emulsions.

### 3.5. Interfacial Tension

Figure 5 and Table 6 show the interfacial tension curve after blended the HYW emulsion and SE emulsion in different proportions. It can be seen that when the content of HYW is 100%, it has the lowest interfacial tension, and the interfacial tension is 6 mN/m after 120 min; with the decrease of HYW content, the interfacial tension increases gradually. When the HYW content is 0%, the interfacial tension reaches 30 mN/m. Take HYW content as 70% to study the effect of demulsifier concentration on interfacial tension, as shown in Figure 4b. It can be seen that with the increase of demulsifier concentration, the interfacial tension also increases rapidly, but the maximum increase is at 100 mg/L. Then increase the concentration of demulsifier, and the interfacial tension increases slightly.

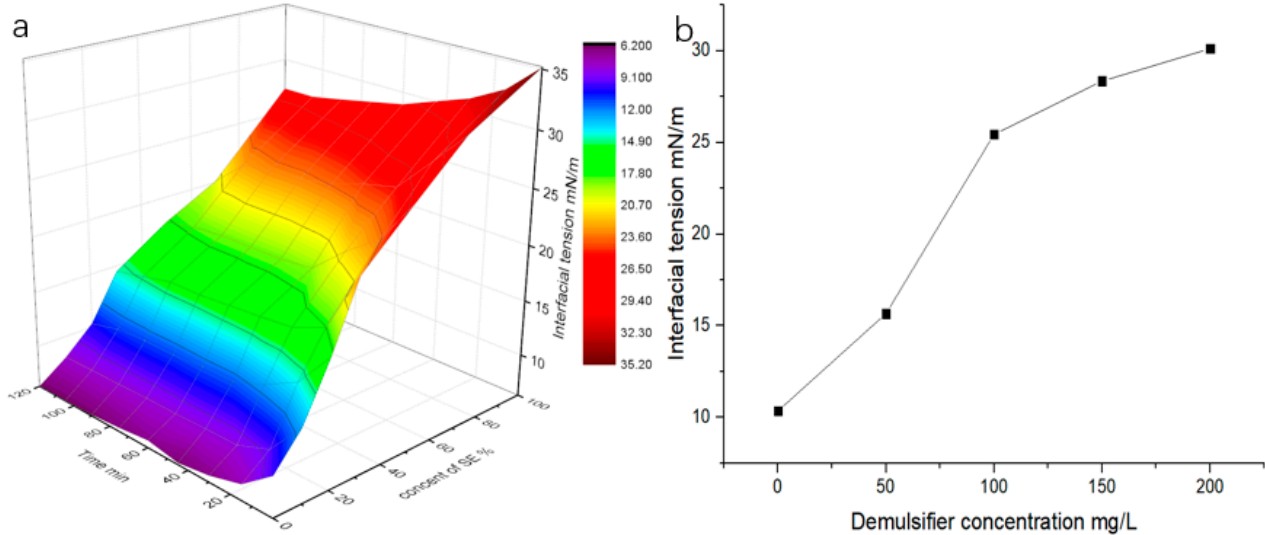

**Figure 5.** (**a**). Effect of SE concentration on oil–water interfacial tension; (**b**). Effect of demulsifier concentration on oil–water interfacial tension of mixed emulsion when SE content is 30%.

**Table 6.** Effect of SE concentration on oil–water interfacial tension.

| Content of SE % | 1 min | 15 min | 30 min | 45 min | 60 min | 75 min | 90 min | 105 min | 120 min |
|---|---|---|---|---|---|---|---|---|---|
| 0 | 9.76 | 7.59 | 6.78 | 6.35 | 6.86 | 6.53 | 6.31 | 6.22 | 6.22 |
| 10 | 12.65 | 9.45 | 8.33 | 8.31 | 8.22 | 8.22 | 8.14 | 8.11 | 8.04 |
| 20 | 17.45 | 14.13 | 12.54 | 11.45 | 11.45 | 11.21 | 11.21 | 11.03 | 10.34 |
| 30 | 23.11 | 17.78 | 16.54 | 16.23 | 16.01 | 15.76 | 15.76 | 15.11 | 14.48 |
| 50 | 27.34 | 23.43 | 21.33 | 18.77 | 18.45 | 18.45 | 18.45 | 18.15 | 17.67 |
| 70 | 31.44 | 28.44 | 25.33 | 24.66 | 23.67 | 23.19 | 23.19 | 22.55 | 20.11 |
| 100 | 35.15 | 32.63 | 31.05 | 30.00 | 28.93 | 28.34 | 27.76 | 27.22 | 27.21 |

### 3.6. Zeta Potential

Figure 6 shows the Zeta potential curve after blended different proportions from HYW emulsion and SE emulsion. It can be seen that when the HYW content is 100%, the absolute value of Zeta potential has the highest absolute value, reaching $-95$ mV. With the decrease of HYW content, the absolute value of Zeta potential gradually decreases. When the HYW content is 70%, the absolute value of Zeta potential decreases the most.

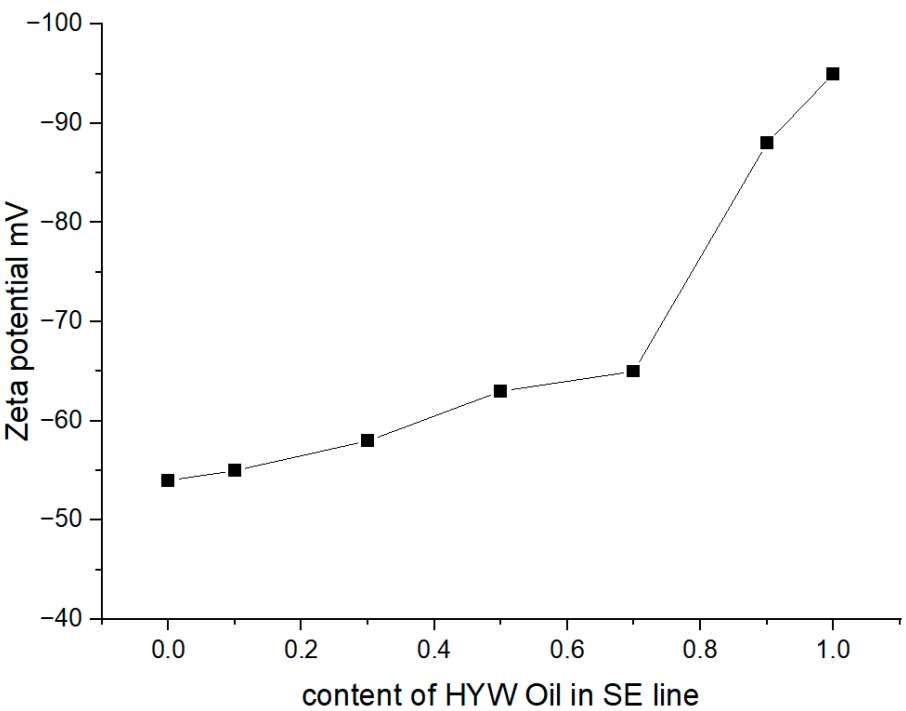

**Figure 6.** Effect of HYW and SE Blended Ratio on Zeta Potential.

## 4. Discussion

### 4.1. Water Separation Rate

According to the requirements of the heavy oil treatment station of Xinjiang Oilfield, the optimal scheme is screened: (1) the 2-h water separation rate in the laboratory reaches 90%; (2) The liquid flow from the heavy oil treatment station will fluctuate occasionally, so the limit proportion cannot be selected, and the liquid flow from the SE line is limited, so the proportion of the liquid flow from the SE line cannot be too large; (3) According to the principle of "improving quality and efficiency" of Xinjiang Oilfield, the treatment temperature should not exceed 70 °C and the concentration of demulsifier should be controlled within 200 mg/L. Based on the above experiments, the optimal demulsification ratio is determined as 70% HYW emulsion content and 30% SE emulsion content. The water separation under this condition is studied, as shown in Figure 6.

Figure 7 shows the change of water separation rate under different temperature and different concentration of demulsifier after blended at 7:3. It can be seen that, when the temperature in the laboratory is 70 °C and the concentration of demulsifier is 120 mg/L, the 2-h water separation rate reaches 90%, which is the optimal condition.

### 4.2. Oilfield Test

On 28 December 2019, the crude oil treatment process was adjusted and optimized. The HYW emulsion and SE emulsion was mixed into No. 4 settling tank for treatment. The demulsification temperature was raised to 70 °C and the demulsifier was reduced to 120 mg/L. After the implementation of this process, the effluent of the first stage is qualified, and the water content is reduced from 20% to 3%, which proves that this demulsification and water separation method is relatively reliable.

The price of demulsifier is RMB 15,000 /ton, and the heavy oil treatment station processes 5000 tons of emulsion every day. When the concentration of demulsifier decreases from 260 mg/L to 120 mg/L, the demulsifier cost will be saved by RMB 2.1 million in 200 days. In addition, when the heating temperature drops from 90 °C to 70 °C, the operating cost of the boiler also drops significantly.

In the future, we will study different types of demulsifiers and biological methods for demulsification of heavy oil emulsion.

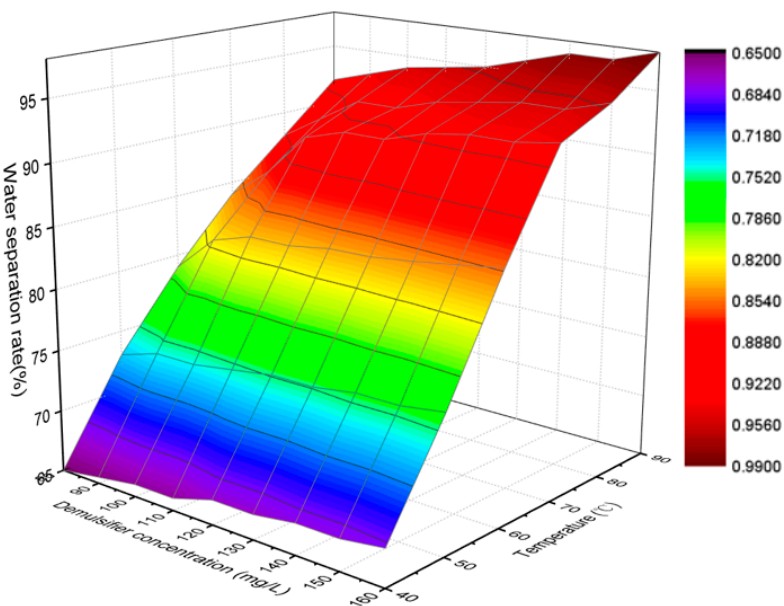

**Figure 7.** Effect of Demulsification Temperature and Demulsifier Concentration on Water fraction.

## 5. Conclusions

In conclusion, the demulsification effect of blended method on HYX heavy oil emulsion was studied in Xinjiang Oilfield. The results show that blended ratio, temperature and centration of demulsifier are the keys on the demulsification processes. When the temperature is 70 °C, the concentration of demulsifier is 120 mg/L, and the blended ratio is 70%, HYW heavy oil emulsion has the best economic demulsification process.

**Author Contributions:** Conceptualization, J.Z.; data curation, Y.P.; funding acquisition, J.H.; investigation, J.C.; methodology, A.A.; software, B.Z.; supervision, J.H.; visualization, J.C.; writing—review & editing, J.Z. All authors have read and agreed to the published version of the manuscript.

**Funding:** This research was funded by [Tianshan Young Scholars] grant number [2018Q031] and [Educational Foundation of Xinjiang Province] grant number [XJEDU2018Y060].

**Conflicts of Interest:** The authors declare no conflict of interest.

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
