# Peer review of "Study on Demulsification Technology of Heavy Oil Blended in Xinjiang Oilfield"

_processes, doi:10.3390/pr11020409_

Round 1

Reviewer 1 Report

The paper is well written. I recommend it for publication 

Author Response

Thank you very much for the reviewer

Reviewer 2 Report

Journal: Processes

Manuscript ID: -

Manuscript Type: Research Article

Manuscript name: Study on Demulsification Technology of Heavy Oil Mixing in Xinjiang Oilfield

The authors investigated HYW heavy oil in Xinjiang Oilfield. The demulsification effect of the SE oil mixed with HYW was studied. Mixing behaviors were determined by applying various tests. It was determined size and type of emulsion, density, viscosity distribution of HYW and SE mixed in different proportions at different temperatures, Zeta potential, interfacial tension and water separation rate. According to the analysis result, Not only the heating temperature drops from 90 ℃ to 70 ℃, the operating cost of the boiler drops significantly but also the demulsifier cost will be saved. The study contributed as a low-cost heavy oil demulsification method.

Considering the above-aforementioned manuscript, the following query should be considered for the manuscript to be comprehensible by the authors.

Query and advise

1-     It should be explained in which regions and in which areas the materials were used before, and information should be given about their economic life.

2-     Where were the materials used in the experimental study obtained, and where were the experimental studies carried out? The necessary explanation should be given.

3-     How was mixing done to ensure homogeneity? (top, middle, bottom).

4-     Flowchart must be demonstrated in the manuscript.

5-     An explanation of the abbreviations should be given.

6-     The test types applied and the number of test repetitions should be given as tables.

7-     The materials used in the studies should be presented as a table with citations.

8-     Future work should be included.

9-     The usage areas of the materials should be specified.

10- Repeated sentences should be avoided.

11- There is no need to use the relevant heading in 2.2. It should be removed from the article.

12- The place of use of formula 1 in the manuscript should be changed.

13- Why were performance tests on the low temperature behavior of the material not carried out?

14- Why weren't experimental studies on the fatigue performance of the material done?

15- What is the chemical property of the materials?

16- Test studies and devices used should be given as a table.

17- What are the future studies? It should be added to the article.

18- What is the material performance against environmental factors?

19- What are the standards used in experimental studies?

20- The experimental study results presented in the tables should be presented with the standards. In addition, it should be stated within which limits or not.

21- The viscosity values in Table 1 should be included in the test results at different temperatures.

22- The work done in the result part should be clarified and the table should be presented after the explanation.

23- In the result part, it should be explained where the differences in the test results between HYW and Se in Table 1 originate.

24- Figure 1 should be rearranged. In the color scale, it should be clarified to which material the results belong. Legend should be used. Max and min standard limits should be shown and units should be written.

25- Threshold values should be given in the figures.

26- What is the reason for the sudden drop in Figure 3d? An explanation must be provided.

27- In Figure 4, it should be stated what the ratios in the legend. It should be rearranged.

28- Units should be given in Figure 5.

29- Titles should not start with figures and tables, and figures and tables should be included after the explanation is made.

30- Figure 6 is not legible, it should be rearranged and the function equation should be added.

31- Current similar studies on the subject of the study should be presented as a table.

32- The conclusion part is not satisfactory. Since it is insufficient, it should be reviewed and detailed.

The manuscript should be rearranged considering the above query and recommendation. It can be better simplified to clear up the confusion. Significant corrections are required. Therefore, The manuscript is not suitable in its current form.

Author Response

  • It should be explained in which regions and in which areas the materials were used before, and information should be given about their economic life.

Reply: Thank the reviewers for their suggestions, the regions, areas and information has been added to the revised paper.

  • Where were the materials used in the experimental study obtained, and where were the experimental studies carried out? The necessary explanation should be given.

Reply: Thank the reviewers for their suggestions, the materials and experimental explanation has been added to the revised paper.

  • How was mixing done to ensure homogeneity? (top, middle, bottom).

Reply: Thank the reviewers for their suggestions, For mixed samples, we shall implement the oil and gas industry standard SY/T 5797-93 of the People's Republic of China. For the original emulsion, we use the middle of the sample to maintain homogeneity..

  • Flowchart must be demonstrated in the manuscript.

The flowchart has been added to the revised paper (Figure 1).

  • An explanation of the abbreviations should be given.

The explanation of the abbreviations has been added to the revised paper.

  • The test types applied and the number of test repetitions should be given as tables.

The test types applied has been added to the revised paper.

  • The materials used in the studies should be presented as a table with citations.

The table has been added to the revised paper.

  • Future work should be included.

The future work has been added to the revised paper

  • The usage areas of the materials should be specified.

This process is currently applicable to the First Oil Production Plant of Xinjiang Oilfield. Whether it is suitable for other heavy oil emulsions needs further study.

  • Repeated sentences should be avoided.

The repeated sentence has been deleted.

  • There is no need to use the relevant heading in 2.2. It should be removed from the article.

It has been revised according to the requirements of the reviewer

  • The place of use of formula 1 in the manuscript should be changed.

It has been revised according to the requirements of the reviewer.

  • Why were performance tests on the low temperature behavior of the material not carried out?

The emulsion viscosity is too high at low temperature to be mixed.

  • Why weren't experimental studies on the fatigue performance of the material done?

Oil field production needs fast demulsification, so it is of no practical significance to study long-term fatigue performance.

  • What is the chemical property of the materials?

The chemical property of the heavy oil emulsion has been added to the revised paper.

  • Test studies and devices used should be given as a table.

Test studies has been added in the flowchart.

  • What are the future studies? It should be added to the article.

The future studies has been added to the revised paper.

  • What is the material performance against environmental factors?

The environment has no impact on the material performance.

  • What are the standards used in experimental studies?

We shall implement the oil and gas industry standard SY/T 5797-93 of the People's Republic of China.

  • The experimental study results presented in the tables should be presented with the standards. In addition, it should be stated within which limits or not.

It has been revised according to the requirements of the reviewer

  • The viscosity values in Table 1 should be included in the test results at different temperatures.

The viscosity of HYW and SE at different temperatures has been shown in Table 4.

  • The work done in the result part should be clarified and the table should be presented after the explanation.

It has been revised according to the requirements of the reviewer

  • In the result part, it should be explained where the differences in the test results between HYW and Se in Table 1 originate.

HYW emulsion is the highly viscous oil emulsion we want to treat. SE emulsion is a low viscosity oil that we find to mix with HYW for viscosity reduction and demulsification.

  • Figure 1 should be rearranged. In the color scale, it should be clarified to which material the results belong. Legend should be used. Max and min standard limits should be shown and units should be written.

It has been revised according to the requirements of the reviewer.

  • Threshold values should be given in the figures.

It has been revised according to the requirements of the reviewer

  • What is the reason for the sudden drop in Figure 3d? An explanation must be provided.

70% of HYW is the phase transition point of the mixed solution. When it is more than 70%, the mixed emulsion shows the characteristics of small particle size of water-in-oil type.

  • In Figure 4, it should be stated what the ratios in the legend. It should be rearranged.

It has been revised according to the requirements of the reviewer

  • Units should be given in Figure 5.

Units mV has been added to the revised paper

  • Titles should not start with figures and tables, and figures and tables should be included after the explanation is made.

It has been revised according to the requirements of the reviewer

  • Figure 6 is not legible, it should be rearranged and the function equation should be added.

Figure 6 is the final experimental result, not the equation fitting.

  • Current similar studies on the subject of the study should be presented as a table.

It has been revised according to the requirements of the reviewer (Table 1).

32- The conclusion part is not satisfactory. Since it is insufficient, it should be reviewed and detailed.

    It has been revised according to the requirements of the reviewer.

Reviewer 3 Report

In this manuscript authors studied the demulsification of heavy oil mixing in a selected oilfield. The topic the research is very important especially in today’s climate where more important oil and gas research is required to provide effective decarbonization. The manuscript is however lacks coherence between the experiments and methods.

Some suggested revisions before the publication such as

1.       Introduction and abstract need to be rewritten based on previous research and significance of the article.

2.       Reason of the temp dependency on density (Fig 2) is not clearly stated

3.       Labeling is incorrect in Fig 3.

4.       Discussion needs to be rewritten on what can be concluded from the experiments

Author Response

  1. Introduction and abstract need to be rewritten based on previous research and significance of the article.

Thank the reviewers for their suggestions, the introduction and abstract has been rewritten in the revised paper.

  1. Reason of the temp dependency on density (Fig 2) is not clearly stated.

The reason of the temp dependency on density has been added to the revised paper.

  1. Labeling is incorrect in Fig 3.

It has been revised according to the requirements of the reviewer.

  1. Discussion needs to be rewritten on what can be concluded from the experiments

It has been revised according to the requirements of the reviewer.

Reviewer 4 Report

1.  Heave oil mixing? (in Keywords) does not seem right

2. The abstract is not written carefully. Claims such as this 'What technology has laid a foundation for research?', 'When the temperature is raised to 90 C and the conc of demulsifier is 260 mg/L, demulsification will not occur?' are not at all clear. Needs a careful re-writing.

3. Introduction - what is HYW? is there an abbreviation table which can be included.

4. The introduction needs a careful rewriting - what is this heavy oil - significance of it - why one wants to create an emulsion - what is the challenge - what are the state of the art solutions - why a solution is being developed. This train of thought could possibly help. This current write-up seems very misleading.

5. Section 3.2 is viscosity temperature curve & I see composition & viscosity. This does not seem right. 

6. What is SE? Need a abbreviation table here again

This has the potential to be a good paper with relevant findings on viscosity reduction. However, the presentation and writing is poor.

Author Response

  1. Heave oil mixing? (in Keywords) does not seem right

Thank the reviewers for their suggestions, the keyword has been changed in the revised paper.

  1. The abstract is not written carefully. Claims such as this 'What technology has laid a foundation for research?', 'When the temperature is raised to 90 C and the conc of demulsifier is 260 mg/L, demulsification will not occur?' are not at all clear. Needs a careful re-writing.

The abstract has been rewritten in the revised paper.

  1. Introduction - what is HYW? is there an abbreviation table which can be included.

All of the abbreviations have been explained in the revised paper.

  1. The introduction needs a careful rewriting - what is this heavy oil - significance of it - why one wants to create an emulsion - what is the challenge - what are the state of the art solutions - why a solution is being developed. This train of thought could possibly help. This current write-up seems very misleading.

The introduction has been rewritten in the revised paper.

  1. Section 3.2 is viscosity temperature curve & I see composition & viscosity. This does not seem right. 

The Section 3.2 has been changed to the “Blended ratio, temperature and viscosity distribution diagram”

  1. What is SE? Need a abbreviation table here again

All of the abbreviations have been explained in the revised paper.

Round 2

Reviewer 2 Report

Although the answers to some questions are not clear, the results are satisfactory.

The article is suitable in its current form. 

Reviewer 3 Report

I agree with the revised version and no further edits are necessary. 

Reviewer 4 Report

NA